# COVID-19: Physical Activity and Quality of Life in a Sample of Swiss School Children during and after the First Stay-at-Home

**DOI:** 10.3390/ijerph19042231

**Published:** 2022-02-16

**Authors:** Valentin Benzing, Patrice Gaillard, David Scheidegger, Alain Dössegger, Claudio R. Nigg, Mirko Schmidt

**Affiliations:** 1Institute of Sport Science, University of Bern, 3012 Bern, Switzerland; patrice.gaillard@students.unibe.ch (P.G.); david.scheidegger@students.unibe.ch (D.S.); claudio.nigg@unibe.ch (C.R.N.); mirko.schmidt@unibe.ch (M.S.); 2Swiss Federal Institute of Sport Magglingen (SFISM), 2532 Magglingen, Switzerland; alain.doessegger@baspo.admin.ch

**Keywords:** physical exercise, well-being, lockdown, Corona pandemic, school children

## Abstract

COVID-19 restrictions and the instructions to stay at home (SaH) may have had an impact on child behavior including physical activity (PA) and health-related quality of life (HRQoL) in Switzerland. Therefore, we investigated PA and HRQoL during and after the SaH in 57 Swiss school children aged 7 to 12 years (*M* = 10.44; *SD* = 1.34). PA was measured using accelerometry (Actigraph GT3X) and HRQoL using the Kid-KINDL^R^ questionnaire. During and post data was compared using paired sample *t*-tests. Independent *t*-tests were used to compare the HRQoL of physically active children with non-physically active children. PA in light (*d* = −0.56), moderate (*d* = −0.44), moderate-to-vigorous (*d* = −0.28) as well as overall HRQoL (*d* = −0.66), psychological well-being (*d* = −0.48), self-esteem (*d* = −0.39), friends (*d* = −0.70) and everyday functioning (*d* = −0.44), were significantly lower during SaH than afterwards. Children who adhered to PA recommendations (+60 min of moderate-to-vigorous PA) during SaH had a better overall HRQoL (*d* = 0.61) and psychological well-being (*d* = 0.56) than those who did not. Since PA levels and HRQoL were lower during SaH compared to afterwards, it seems that the restrictions negatively impacted children’s PA and HRQoL. During future SaHs, promoting children’s PA and HRQoL seems important.

## 1. Introduction

On 11 March 2020, the Director-General of the World Health Organization (WHO) declared an outbreak of the COVID-19 pandemic [1]. On 16 March 2020, the Swiss Federal Council introduced stringent measures. All non-essential businesses and services including shops, restaurants, bars, entertainment and leisure facilities were closed immediately. Meetings of more than five persons, face-to-face teaching at all educational institutions, public as well as private events, including sporting events and club trainings, were banned until 11 May 2020 [2]. At the same time, governments worldwide imposed restrictive measures that massively curtailed public and private life. An initial stay-at-home (SaH) period was posted in a large number of countries to stop the rapid increase in COVID-19 incidence rates [3]. In Switzerland, the instruction was to “stay at home” and “only to go out of the house if absolutely necessary”. Although necessary to reduce the incidence rate [4], it is possible that restrictions (e.g., closed schools, sports clubs) and possible related insecurities (e.g., the fear of contracting the virus) may have also reduced physical activity (PA) of school children.

PA is defined as all energy-consuming bodily movements produced by skeletal muscles [5]. The WHO recommends on average 60 min of moderate-to-vigorous PA (MVPA) per day for children [6]. The positive effects of regular PA on health in children and adolescents are well- known [7,8] and children’s PA has not only been linked to physical but also mental health and well-being [9,10,11,12].

For example, cross sectional and interventional studies indicate that PA is associated with, and has a positive effect on, health-related quality of life (HRQoL; [11,13]). HRQoL refers to a multidimensional construct including physical, mental, social, psychological and functional aspects of well-being [14,15]. Interacting neurobiological, psychosocial, behavioral mechanisms are hypothesized to underlie the relationship between PA and mental health including well-being [9,16].

Concerning the effects of the COVID-19 pandemic on school children’s PA and HRQoL, researchers assumed that restrictions could lead to reduced PA and increased sedentary behavior, subsequently posing a threat to (mental) health [17,18,19]. Indeed, when looking at mental health and HRQoL of children, studies report lower mental health and HRQoL during SaH [20,21,22]. Studies from different countries (e.g., China, USA) found that children during SaH are worried, have higher levels of stress, anxiety and depressive symptoms and worsened psychological well-being [21,23,24,25,26,27]. A representative survey study using a sample of 1586 German families with 7-to 17-year-old youths reports a significantly lower HRQoL in youth than before the pandemic [28].

Similarly with regard to PA, most available studies (e.g., Canada, China, Croatia, Italy, Spain, USA) on school children (exclusively using subjective assessments of PA) found (partly) substantial decreases in PA [29,30,31,32,33,34,35]. However, a study from Germany using a representative sample of 1711 4- to 17-year-olds found a decrease in organized PA, but an increase in total PA, which was also reflected in a larger number of children adhering to the WHO guidelines on PA during SaH compared to before [36]. However, most studies used self-reports and not much is known on objective assessments of PA during SaH.

To our knowledge, this is the first study investigating children’s PA objectively during the COVID-19 SaH in a longitudinal study design. In the current study, we quantified PA of 7–12-year-old children in Switzerland during and after the initial SaH using accelerometry. In addition, HRQoL was assessed by questionnaire, making this one of the very few available studies combining objectively measured PA and HRQoL in children during this extraordinary situation. Based on the available empirical evidence, we hypothesized a reduction in PA during SaH compared to afterwards. Based on the available reports on the effects of the COVID-19 pandemic on children´s mental health, we hypothesized to find lower HRQoL during SaH compared to afterwards. Further, we hypothesized that children spending on average at least 60 min in MVPA have a higher HRQoL compared to those not meeting the PA guidelines. Further, we explored associations between change in PA (during–after SaH) with background variables.

## 2. Materials and Methods

### 2.1. Design and Procedure

The current study was conducted in the Swiss cantons of Zurich and St. Gallen using a repeated measures (two timepoints during and after SaH) design. At two timepoints, PA behavior (accelerometry) and a survey including background variables and HRQoL were assessed. The first measurement took place during the initial SaH from 21 April to 4 May 2020, a second measurement (24 June to 3 July 2020) was conducted after restrictions were substantially reduced (for the chronological order of the COVID-19 pandemic, the stringency of restrictions in Switzerland and the timeframes of the current study’s assessments see Figure 1; data for incidence rates and government stringency index were available online: https://ourworldindata.org/grapher/covid-stringency-index (accessed on 14 January 2022). Notably, at the time of the second measurement, the “extraordinary situation” was lifted again. This meant that only large events with over 1000 people remained banned and hygiene measures remained in place (no handshaking, regular handwashing). All other restrictions were discontinued. Thus, schools and sports clubs were allowed to resume including contact sports, tournaments and competitions. In addition, all stores, services as well as recreational facilities were opened again.

### 2.2. Participants

Due to the difficult situation during the first SaH, two aspects concerning the sample have to be considered: First, the sample of the current study consists of a convenience sample. A representative sample was not possible to assess due to the little time we had at our disposal during the first SaH. Second, no a priori power calculation was performed since the study was planned in a very short period and our goal was to include as many children as possible.

In total, 70 children were recruited at three different schools for the current study. In a first step, eligible participants were informed about the study by their teachers. Subsequently, the research team contacted each family and informed about the study via phone resulting in a 95% participation rate.

During the first measurement point, out of the 70 children, nine discontinued participation due to: accident or illness; overload of parents due to the general situation around the pandemic; child did not want to wear the accelerometer anymore or it was forgotten to be put on several times. In addition, four children had not enough valid wear time in accelerometry data (at least ten hours on at least three weekdays and eight hours on at least one weekend day; [37,38]). This resulted in a sample of 57 children for the first time point (see Table 1 for background characteristics).

Prior the second measurement point, out of the 57 remaining children, eleven declined participation due to: (a) accident or illness; (b) child did not want to wear the accelerometer (at school) or it was forgotten to be put on several times; (c) children indicated that they associate the first measurement with a negative time (pandemic) and now look forward to a time without those negative feelings. During the second measurement point another four children discontinued participation due to the following reasons: (a) they did not want to wear or forgot to wear the accelerometer; (b) accident or illness. In addition, six children did not have enough valid wear time during the second measurement. This resulted in 36 participants for the second time point.

When comparing participants (using independent *t*-tests) with data at both time-points with those participating at the first time-point only (see Appendix A), significant differences were found in socioeconomic status. Given this result missing at random (MAR) was assumed indicating the use of multiple imputation methods including (among others) socioeconomic status as predictor. Multiple imputation methods are superior to other approaches such as listwise deletion (also in small samples; [39]). Therefore, data were imputed applying fully conditional specification (predictive mean matching). Fully conditional specification was based on all available variables of the dataset and all missing dependent variables were imputed. Overall pattern of results was similar when using original or multiple imputed data for analyses. Given the superiority of multiple imputation methods and the higher statistical power compared to list-wise deletion [39], results report pooled estimates derived from the analyses conducted with the multiple imputed data set.

### 2.3. Measurements

Background variables. General information on children (age, sex, height, weight, class, nationality, membership in sports clubs) was obtained. Body mass index (BMI) was calculated using the Center for Disease Control and Prevention (CDC) calculator; overweight was determined based on percentile ranks of the CDC growth charts, which are age and sex specific [40]. Socioeconomic status was measured using the Family Affluence Scale III (FAS III; [41,42]). The instrument consists of six items that ask, for example, about the number of vehicles in the household or the number of family vacations in the past year. For each item between 0 and 3 points may be obtained with the highest total score of 13. Studies have provided evidence for the validity of the FAS III [41] and found that the FAS III correlates with parent-reported income [43]. In addition, questions on the living conditions were derived from the Australian CLAN- and the Swiss SCARPOL study [44,45]. These items were assessed in the SOPHYA study (a large cohort study on PA in Swiss children [46,47]) and therefore were also included in the current study.

PA behavior. Accelerometers (GT3x, Actigraph, Pensacola, Florida) measured body accelerations on three axes. Accelerometry has the advantage to provide a dense assessment of PA over a longer time-period covering also short PA bursts, which are more frequent in children [37,48]. Devices were worn for ten consecutive days on the right wrist and only removed for sleeping, showering, or swimming. If the device was taken off, this was documented in a diary.

Initialization (epoch length 15 s), downloading, validity checking, and analysis were performed using Actilife software (Actilife 6.12, Pensacola, FL, USA). Data were considered valid if devices were worn for at least ten hours on at least three weekdays and for eight hours on at least one weekend day [37,38]. The activity was classified as sedentary, light, moderate, or vigorous intensity. In accordance with the SOPHYA-study [46,47], the following cutoffs based on Freedson et al. [49] were chosen for classification: Metabolic equivalent of tasks (METs) = 2.757 + (0.0015 × counts per minute) − (0.08957 × age (in years)) − (0.000038 × counts per minute (cpm) × age (in years)). To this end, the threshold for MVPA is 4 METs. Accordingly, for a 10-year-old participant, <100 cpm is considered sedentary or supine (1–1.5 METs), 100–1909 cpm is considered light PA (1.5–4 METs), 1910–3695 cpm is considered moderate PA, (4–6 METs), and greater than 3696 cpm is considered vigorous PA (>6 METs).

Health related quality of life. HRQoL was assessed using the parent version of the Kid-KINDL^R^ (see for further information Available online: https://www.kindl.org (accessed on 14 February 2022); [15]). The questionnaire includes 24 items which are rated on a 5-point Likert scale (never, seldom, sometimes, often, always). The items cover the following six dimensions of HRQoL over the past week: physical well-being, psychological well-being, self-esteem, family, friends, and everyday functioning (school) and a total score (average of the six dimensions). Raw scores are transformed to range from 0 to 100, with higher scores indicating high quality of life. The Kid-KINDL^R^ is a validated instrument with acceptable psychometric properties (reliability: Cronbach’s alpha = 0.85; [50,51]).

### 2.4. Statistical Analyses

Statistical tests were performed using SPSS 27.0 (SPSS Inc., Chicago, IL, USA). First, the total accelerometry wear time was compared between the first and second measurement point using paired sample *t*-tests. Because we did not find statistically significant differences (*p* > 0.05), unadjusted paired sample *t*-tests were used for the comparisons of PA and HRQoL during SaH with after SaH. To compare the prevalence of children reaching at least 60 min of MVPA during SaH with after SaH, a (paired) binominal McNemar test was used. Second, to further investigate HRQoL in active vs. inactive children during SaH, independent *t*-tests were used (note that results report Welch *t*-tests in case of unequal variances or sample sizes). Therefore, children who fulfilled PA guidelines (60+ minutes of MVPA per day) were compared to children who did not fulfill the PA guidelines on all HRQoL scales. Third, to explore potential moderating variables, associations of background variables with change in PA levels and HRQoL Pearson correlations were calculated. Therefore, change scores (during SaH–after SaH) were calculated for minutes per day in sedentary, light, moderate, vigorous PA as well as for the scales of the KINDL^R^ (physical well-being, psychological well-being, self-esteem, family, friends, and everyday functioning) and correlated to age, sex, BMI, socioeconomic status. In addition, to explore whether change in PA was related to change in HRQoL, Pearson correlations were calculated. Given the exploratory fashion of these correlational analyses, no correction for multiple testing was applied. For all analyses, Cohen’s *d* was reported as an estimation of effect size (small effect size = 0.2, medium effect size = 0.5, large effect size = 0.8) and the level of significance was set a priori at *p* < 0.05. Note that descriptive statistics (e.g., in Table 1) include the averages from the multiple imputed datasets.

## 3. Results

### 3.1. PA and HRQoL during SaH Compared to after SaH

Accelerometry data showed that sedentary time increased (*d* = 0.45) and time spent in light (*d* = −0.56), moderate (*d* = −0.44) and MVPA (*d* = −0.28) was reduced from during SaH compared to after the SaH (see Table 2). In addition, during SaH less children (56.1% compared to 77.2% after SaH) fulfilled WHO PA recommendations (*p* = 0.023).

In HRQoL, lower values were observed in the total (*d* = −0.66), the psychological well-being (*d* = −0.48), friends (*d* = −0.70) and everyday functioning score (*d* = −0.44) during SaH compared to after SaH (see Table 2).

### 3.2. HRQoL in Active vs. Inactive Children during SaH

As can be seen in Table 3, children who were physically active for on average at least 60 min daily during SaH had a better HRQoL in the total score (*d* = 0.61) and the psychological well-being score (*d* = 0.56). In addition, small to moderate effect sizes were observed in the self-esteem (*d* = 0.47) and the everyday functioning scale (*d* = 0.42).

### 3.3. Exploratory Analyses

With regard to potential moderating variables of PA behavior, Pearson correlations between ΔPA (Δsedentary time, Δlight PA, Δmoderate PA, ΔMVPA, Δvigorous PA) with background variables (age, sex, BMI, socioeconomic status), showed statistically significant associations for: sex with Δsedentary time (*r* = –0.447, *p* < 0.001) and Δlight PA (*r* = 0.468, *p* < 0.001) and age with Δmoderate PA (*r* = −0.301, *p* = 0.025). No other statistically significant associations were found (*p*s > 0.05).

With regard to potential moderating variables of HRQoL, Pearson correlations between ΔHRQoL (Δtotal score, Δphysical well-being, Δpsychological well-being, Δself-esteem, Δfamily, Δfriends, Δeveryday functioning) with background variables (age, sex, BMI, socioeconomic status) showed statistically significant associations for: age with Δeveryday functioning (*r* = −0.277, *p* = 0.041), sex with Δtotal score (*r* = 0.266, *p* = 0.047), BMI with Δtotal score (*r* = –0.334, *p* = 0.013; Appendix A), Δphysical well-being (*r* = −0.346, *p* = 0.009), Δself-esteem (*r* = −0.369, *p* = 0.005) and Δeveryday functioning (*r* = −0.336, *p* = 0.013). No other statistically significant associations were found (*p*s > 0.05).

With regard to the potential associations between change in PA and change in HRQoL, no statistically significant correlations were found (*p*s > 0.05).

## 4. Discussion

In the current study, we investigated PA and HRQoL in children during and after the first COVID-19 related SaH in Switzerland. The study revealed three main results. First, results show lower PA and higher sedentary time during SaH than afterwards in our sample of Swiss school children. Second, in these children HRQoL was lower during SaH than afterwards. Third, children who were on average at least 60 min physically active per day had a better HRQoL than children who were less active during SaH.

We observed less time spent in light, moderate, MVPA and an increase in sedentary time during SaH compared to afterwards. Related to this finding, fewer children have been able to meet the recommendations of on average at least 60 min of MVPA per day during SaH. In the WHO PA guidelines [6,52], PA was found to be associated with a variety of health outcomes including physical fitness, cardiometabolic health, bone health, cognitive outcomes, reduced obesity and mental health. In addition, more time spent sedentary was found to be associated with negative health effects on fitness, cardiometabolic health, adiposity, pro-social behavior, and sleep duration. Although necessary to reduce incidence rates [4], we assume that more stringent restrictions during SaH have led to more inactivity in our sample (compared to afterwards), bearing a potential risk for negative health effects.

The findings of lower PA levels during SaH than afterwards are in line with the majority of studies using questionnaire data in children and adolescents [29,30,31,32,33,34,35]. For example, studies from China and Spain found that PA reduced by 81.6% and 51.7%, respectively. Furthermore, in China, sedentary time during SaH nearly tripled. In the present study, however, children spent only 9.2% less in MVPA during SaH and only 28 min more sedentary than afterwards. However, when comparing the results of the current study to other studies, one must consider two major differences between these studies. First, in the current study PA and HRQoL during SaH were compared to shortly afterwards, when restrictions were eased. Thus, there may be residual effects of the SaH reflected in lower PA levels. Second, in the current study, PA was assessed using accelerometry instead of retrospective questionnaire data. Although, these differences might have influenced the results, it seems that the stringency of the governmental restrictions (see Appendix A for comparison of stringency between Switzerland, Spain and China) influenced PA behavior in our sample of Swiss children.

Considering a study in adults [53,54] and a recent representative survey in German children and adolescents [36], both studies found an increase in PA during the first SaH. Given that the restrictions (see Appendix A) in Germany and Switzerland were similar (even slightly more stringent in Germany), this raises the question how these different findings may be explained. Related to the above-mentioned differences, a first potential explanation for contradictory results may be found within the measurement and operationalization of PA and the time point when it was assessed. The self-report questionnaire used in the study by Schmidt et al. [36] was applied before and during SaH. For younger children, it was filled out together with their parents and assessed multiple dimensions of PA including sports activity and habitual PA. Overall, the authors found that sports activity declined, whereas habitual PA increased, resulting in an overall increase in PA during SaH. In contrast, in the current study, time in different PA levels was gathered during and after SaH using accelerometry data. With accelerometry it is difficult to distinguish between habitual and sports activity, however, it is less prone to biases as recall measures are [37]. Biases could arise, for example, from more time spent together between parents and children during SaH, influencing parent ratings of habitual PA due to a higher availability of information (availability heuristic; [55]). Considering the comparable restrictions in both countries, these dissenting findings are probably related to the time of measurement (see limitations), the measurement itself and the different PA constructs reflected.

Previous studies showed that factors such as age or living environment were associated with PA behavior during SaH [30,32]. The exploratory finding of the current study, showing that age was related to change in PA is in line with studies from Canada and Portugal [32,56]. Since in younger children habitual PA accounts for a larger percentage of total PA [36], these children may be less affected by the SaH and the associated ban of organized sports. Furthermore, when considering other determinants of PA derived from the ecological model of PA [57] such as the living environment, it seems that the participants of the current study may have had a low risk for a PA decline during SaH. This is because participants were young, had a comparably high socioeconomic status and lived in PA friendly environments (95% were from rural and suburban areas, 87.7% lived in a pedestrian friendly neighborhood, 96.5% had access to a garden, 89.5% had a playground nearby). Although these exploratory findings of the current study have to be interpreted cautiously, because they were not controlled for multiple testing, it is likely that there are more vulnerable populations (for example children with low socioeconomic status from urban areas) which were even more affected by the initial SaH than the sample included in the current study.

HRQoL was found to be reduced during SaH compared to afterwards in the current study. This finding is in line with previous studies comparing mental health and HRQoL in children before and during SaH [20,21,22]. When comparing the scores for HRQoL after SaH from our sample to the norms from Germany (age 7–10 [50]) we found in the current study higher average values (above the 95% confidence interval) in all scales except for everyday functioning, which was equally high. However, during SaH, children in the current study had lower average values (mean below the 95% confidence interval) in the total, the psychological well-being, the friends and everyday functioning scale compared to the norms. Due to the restrictions including limited meetings with other children, a substantial difference in the friend scale was expected. However, the detected difference in the psychological well-being scale (example questionnaire items: In the last week: my child laughed a lot and had fun–my child felt alone–my child felt anxious or insecure) indicates that the first SaH was generally perceived more negatively than expected.

In addition, we found reduced HRQoL in insufficiently active children (less than 60 min of MVPA per day) during SaH for the overall and the psychological well-being score and a tendency for self-esteem. Although underlying mechanisms remain unclear, this finding may be explained with psychosocial mechanisms [9]. It is assumed that PA provides an opportunity to satisfy basic psychological needs including autonomy, relatedness, and competence [58]. If PA is being reduced it could negatively affect physical self-perceptions (e.g., physical self-concept) and well-being. In addition, because organized sports and PA with other children was restricted during SaH, social support by peers might have also been limited. To speculate, considering the assumptions of the Exercise and Self-Esteem Model [59] and a recent empirical investigation showing that social support (e.g., from peers) is highly important for self-esteem and MVPA [60], restrictions may thus have led (directly and indirectly) to both reduced PA and HRQoL during SaH.

With regard to the explorative analyses on moderating variables, the association between BMI and change in HRQoL deserves further interpretation. In detail, BMI was negatively related to change in several scales of HRQoL including the total score, physical well-being, self-esteem, and everyday functioning. This finding is in line with a previous meta-analysis showing that, in general, BMI and HRQoL are related in children and adolescents [61]. When having a closer look at the correlation (see Appendix A for scatter plot), it becomes apparent that overweight children have a particular influence on the correlation. We therefore speculate, that children with a larger BMI may have been particularly affected by SaH in terms of their psychosocial and physical HRQoL. However, this has to be interpreted cautiously because only seven children were overweight in the current sample.

### Limitations

The following limitations should be considered when interpreting study results. First, one has to be aware that the sample size of the current study was small, and that due to the extraordinary situation, we had a high percentage of missing data. Although multiple imputation methods were used, these circumstances may still be sources of greater bias when imputing missing data. Second, assessments took place during and after SaH and not during and before SaH. Although known from a large representative cohort study from Switzerland (SOPHYA study), that there is no difference in MVPA (measured by accelerometry) between spring and summer (*M_spring_* = 82.9, *M_summer_* = 83.9, *p* > 0.05; [62]), seasonal effects still might have influenced results. Third, the sociodemographic background of the participants was not representative for Switzerland. Notably, PA data after SaH were found to be comparable to the SOPHYA study [62]. For example, similar values for time spent in MVPA were found in the SOPHYA (age groups 8–9 = 100.7 min; 10–11 = 72.5 min; 12–13 = 58.6 min) and the current study (age groups: 8–9 = 93.87 min; 10–11 = 69.61 min; 12–13 = 66.80 min). It nevertheless is possible that the effect of SaH on PA is underestimated as a previous study found the largest reductions in urban areas and the sample of the current study consisted mainly of children from rural areas [30]. Given this selection bias, the generalizability of results to Swiss children in general is not possible. Fourth, sample size was powered for comparisons of PA and HRQoL during and after SaH, but not to detect relationships between change in PA, HRQoL with background variables. Therefore, these analyses remain exploratory and must be interpreted cautiously. Fifth, the current study used only two time points (one assessment during a period of stringent measures and one after). Given the volatile situation, multiple assessments during the pandemic and an additional assessment before the SaH would have been beneficial. Since we have no information on children’s baseline levels of activity, it is possible that after restrictions were eased, they were more active due to being confined at home and indoors for some time. In summary, these limitations may limit generalizability.

## 5. Conclusions

PA and HRQoL were lower during the initial SaH in a sample of Swiss school children compared to afterwards. The difference in PA was smaller compared to countries with very stringent SaH restrictions, however, it is still alarming when considering the relationship of PA with physical and mental health. What we currently don’t know is how future SaHs affect PA behavior and HRQoL, and how persistent the negative effects on PA and health are. Therefore, these issues need to be investigated in future studies. Especially in phases of increased stress, care should be taken to ensure that children are sufficiently active [63]. This could be achieved, for example, by innovative large scale digital PA programs (as currently partly used on a smaller scale) or by increased teachers, trainers and parental awareness and capacity development. Therefore, we suggest that the general recommendations in similar situations should be “stay home, stay safe and stay active!”.

## Figures and Tables

**Figure 1 ijerph-19-02231-f001:**
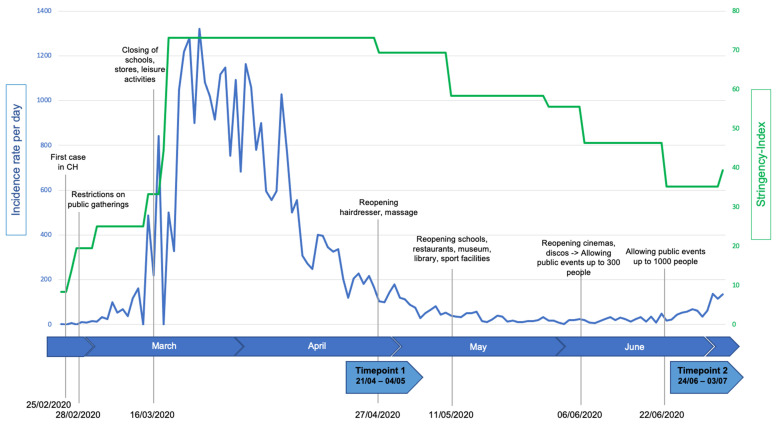
Chronological order of important events, incidence rate and government stringency index related to COVID-19 before and during the time of the study.

**Table 1 ijerph-19-02231-t001:** Background characteristics.

	School Children (*n* = 57)
	Mean (SD)
Age [years]	10.44 (1.38)
Socioeconomic status [0–13]	9.04 (2.00)
Height [cm]	143.43 (17.60)
Weight [kg]	36.97 (8.64)
BMI [kg/m^2^]	17.53 (3.03)
BMI percentiles (CDC growth charts)	45.89 (27.86)
	*n* (%)
Overweight	7 (12.3%)
Sex [female]	34 (59.6%)
Swiss citizenship	57 (100%)
Swiss citizenship parents	54 (94.7%)
School grade	
1st Grade	3 (5.3%)
3rd Grade	14 (24.6%)
4th Grade	8 (14.0%)
5th Grade	22 (38.6%)
6th Grade	10 (17.5%)
Membership in sports club	46 (80.7%)
Living conditions	
Residential area	
Urban	3 (5.3%)
Suburban	8 (14.0%)
Rural	46 (80.7%)
Access to garden	55 (96.5%)
Nearby playground	51 (89.5%)
Pedestrian friendly neighborhood	50 (87.7%)

Note. SD = standard deviation; BMI = body mass index; CDC = Center for Disease Control and Prevention.

**Table 2 ijerph-19-02231-t002:** Physical activity and HRQoL during and after stay-at-home.

	During SaH (*n* = 57)	After SaH (*n* = 57)	95% *CI* of Difference			
	*M (SD)*	*M (SD)*	ΔM	Lower	Upper	*p* (*d*)
**Physical Activity (Accelerometry; Minutes per Day)**						
MVPA	68.53 (28.68)	75.48 (23.14)	−6.95	–0.13.51	−0.40	0.037 * (−0.28)
Sedentary	626.21 (72.70)	598.63 (48.82)	27.58	11.87	43.29	0.001 * (0.45)
Light	228.16 (55.62)	254.98 (29.59)	−26.82	−39.15	−14.49	<0.001 * (−0.56)
Moderate	44.75 (17.93)	51.70 (12.73)	−6.95	−11.12	−2.77	0.001 * (−0.44)
Vigorous	23.91 (14.02)	23.82 (12.48)	0.09	−3.44	3.62	0.958 (0.01)
**HRQoL**						
Total score	76.26 (10.21)	82.96 (8.31)	−6.70	−9.36	−4.03	<0.001 * (−0.66)
Physical well-being	80.15 (16.19)	83.79 (11.01)	−3.63	−8.08	0.82	0.110 (−0.21)
Psychological well-being	80.15 (14.40)	87.89 (11.28)	−7.74	−11.91	−3.57	<0.001 * (−0.48)
Self-esteem	75.22 (13.35)	80.74 (9.41)	−5.52	−9.17	−1.87	0.003 * (−0.39)
Family	79.93 (11.69)	81.47 (9.63)	−1.54	−4.36	1.28	0.285 (−0.15)
Friends	65.17 (21.35)	81.26 (12.07)	−16.09	−22.12	−10.06	0.001 * (−0.70)
Everyday functioning	76.90 (16.15)	82.58 (12.41)	−5.68	−9.16	−2.20	0.002 * (−0.44)

Note. SaH = stay at home; *M* = mean; *SD* = standard deviation; ΔM = difference score (during SaH–after SaH); *CI* = confidence interval; *p* = level of significance; *d* = Cohen’s d; MVPA = moderate-to-vigorous physical activity; HRQoL = health-related quality of life; * indicates *p* < 0.05.

**Table 3 ijerph-19-02231-t003:** Comparison between children fulfilling PA recommendations (60 min of MVPA per day) or not during and after stay-at-home.

	During SaH		95% *CI* of Difference			*p* (*d*)	After SaH		95% *CI* of Difference			
	≥60 min of Daily PA (*n* = 32)	<60 min of Daily PA (*n* = 25)					≥60 min of Daily PA (*n* = 44)	<60 min of Daily PA (*n* = 13)				
	*M (SD)*	*M (SD)*	ΔM	Lower	Upper		*M* (*SD*)	*M* (*SD*)	ΔM	Lower	Upper	*p* (*d*)
**HRQoL**												
Total score	78.91 (9.85)	72.87 (9.64)	6.04	0.85	11.24	0.023 * (0.61)	84.01 (10.20)	83.73 (8.02)	0.28	−5.94	6.50	0.930 (0.03)
Physical well-being	82.62 (16.60)	77.00 (15.08)	5.62	−2.87	14.11	0.195 (0.35)	86.14 (14.16)	81.73 (10.83)	4.41	−4.10	12.92	0.310 (0.33)
Psychological well-being	83.59 (11.67)	75.75 (16.24)	7.84	0.46	15.22	0.037 * (0.56)	91.03 (11.64)	84.62 (14.64)	6.42	−3.18	16.02	0.190 (0.49)
Self-esteem	77.93 (12.60)	71.75 (13.48)	6.18	−0.74	13.10	0.080 (0.47)	81.25 (12.09)	82.21 (9.13)	−0.96	−8.18	6.26	0.794 (−0.08)
Family	81.84 (11.09)	77.50 (12.00)	4.34	−1.78	10.46	0.165 (0.37)	80.71 (11.20)	85.58 (10.23)	−4.87	−12.31	2.57	0.199 (−0.44)
Friends	67.52 (22.18)	62.17 (19.85)	5.35	−5.94	16.63	0.353 (0.25)	80.98 (14.22)	82.69 (13.70)	−1.71	−11.47	8.04	0.731 (−0.12)
Everyday functioning	79.88 (16.00)	73.08 (15.54)	6.80	−1.61	15.21	0.113 (0.42)	83.97 (14.50)	85.58 (10.80)	−1.61	−10.21	6.99	0.714 (−0.12)

Note. SaH = stay at home; PA = physical activity; M = mean; ΔM = Mean difference (during SaH–after SaH); *SD* = standard deviation; *CI*= confidence interval; *p* = *p*-value; *d* = Cohens d; * indicates *p* < 0.05.

## Data Availability

Data is available upon reasonable request.

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
