# Peer review of "COVID-19: Physical Activity and Quality of Life in a Sample of Swiss School Children during and after the First Stay-at-Home"

_ijerph, 2022, doi:10.3390/ijerph19042231_

Round 1
Reviewer 1 Report
The title of the paper and the interpretation of the results suggest that the data are representative for Swiss schoolchildren which can’t be correct based on the number of participants (N=57 at the first time point and N=36 in the second time point), and the fact that the sample is no way representative of the entire population of Swiss children, e.g. most of the children (80.7%) from the sample are from rural areas, and only 5.3% of children included in the research are from urban areas.
Additionally, although the paper describes the use of multiple imputation methods which is explained and justified with a paper investigating the use of imputations in small samples (Kleinke, 2018), the referenced paper clearly states that small sample size and high percentage of missing data (which is the case in this study) can still be sources of greater bias when imputing missing data.
Further, the time period between the two measurements is very short: the first measurement took place during the initial stay-at-home from April 21 to May 4, 2020, and the second measurement took place from June 24 to July 3, 2020 after restrictions were substantially reduced. It is very hard to make any substantial conclusions in such a short time period. If the investigators wanted to investigate if children are more physically active after the lockdown period they should have included additional time points where they would see if the observed effect is indeed present and longlasting. Also, we can not be sure if these children have now returned to their previous levels of activity, pre-COVID measures, since we have no information on their baseline levels of activity. It could be that after stay-at- home orders were lifted children were more active than ever before, for a short period of time, due to being confined at home and indoors for some time.
Considering the big attrition, the small sample size and the lack of a baseline measure, it is hard to make any generalizable conclusions from this study that would be of use to a broader audience.
Author Response
Reviewer 1
First, we would like to thank the reviewer for the careful examination of our manuscript. We sincerely appreciate the important and constructive comments and critique. We have revised the manuscript based on these suggestions and we are sure that it has improved substantially by these changes.
The title of the paper and the interpretation of the results suggest that the data are representative for Swiss schoolchildren which can’t be correct based on the number of participants (N=57 at the first time point and N=36 in the second time point), and the fact that the sample is no way representative of the entire population of Swiss children, e.g. most of the children (80.7%) from the sample are from rural areas, and only 5.3% of children included in the research are from urban areas.
Thank you very much for this comment. Although this issue was included in the limitations section, from this comment we understand that we did not fully succeed in avoiding over-interpretation. Therefore, we firstly added “in a sample of Swiss school children” to the title so that it directly becomes clear that results are not representative. Secondly, we added a note on this issue to the Participants section (see below). Thirdly, we amended the limitations section to make this issue clearer (see lines 374-378). Fourthly, we checked throughout the whole manuscript and amended where appropriate to avoid over-interpretation.
“Due to the difficult situation during the first SaH, two aspects concerning the sample have to be considered: First, the sample of the current study consists of a convenience sample. A representative sample was not possible to assess due to the little time we had at our disposal during the first SaH. Second, no a priori power calculation was performed since the study was planned in a very short period and our goal was to include as many children as possible.”
Additionally, although the paper describes the use of multiple imputation methods which is explained and justified with a paper investigating the use of imputations in small samples (Kleinke, 2018), the referenced paper clearly states that small sample size and high percentage of missing data (which is the case in this study) can still be sources of greater bias when imputing missing data.
This is correct. Therefore, we added this as first issue in our limitation section (see lines 362-365).
“First, one has to be aware that the sample size of the current study was small and that due to the extraordinary situation, we had a high percentage of missing data. Although multiple imputation methods were used, these circumstances may still be sources of greater bias when imputing missing data.”
Further, the time period between the two measurements is very short: the first measurement took place during the initial stay-at-home from April 21 to May 4, 2020, and the second measurement took place from June 24 to July 3, 2020 after restrictions were substantially reduced. It is very hard to make any substantial conclusions in such a short time period. If the investigators wanted to investigate if children are more physically active after the lockdown period they should have included additional time points where they would see if the observed effect is indeed present and longlasting. Also, we can not be sure if these children have now returned to their previous levels of activity, pre-COVID measures, since we have no information on their baseline levels of activity. It could be that after stay-at- home orders were lifted children were more active than ever before, for a short period of time, due to being confined at home and indoors for some time.
Thank you for this comment. Indeed, multiple assessment periods would have been beneficial. Due to the extraordinary situation, unfortunately, this was not possible. We added this issue and your concerns from the next comment to the limitation section (see lines 381-387):
“Fifth, the current study used only two time points (one assessment during a period of stringent measures and one after). Given the volatile situation, multiple assessments during the pandemic and an additional assessment before the SaH would have been beneficial. Since we have no information on children’s baseline levels of activity, it is possible that after restrictions were eased, they were more active due to being confined at home and indoors for some time. In summary, these limitations may limit generalizability.“
Considering the big attrition, the small sample size and the lack of a baseline measure, it is hard to make any generalizable conclusions from this study that would be of use to a broader audience.
Reviewer 2 Report
Line 26: Add the year with the date
Line 43-44 (and throughout the rest of the paper): Why do you provide citations with names/years here but also in the requested format as brackets? I would suggest removing all named citations like this throughout the paper.
Line 79: Grammar issue with (during to after SaH). Please revise.
Line 123: Remove the semicolon and replace with 'or'. Semicolon would not be used in this instance
Line 142: The CDC title should be capitalized
Line 153-157: Not sure you need to list out the exact questions like this. I would suggest removing and provide the citation for the instrument.
Line 161: Can you provide more details regarding the 10 day data collection period? Was this consecutive days or only days when they had school? It also states "during school" - does this include online education if the schools were shut down? Please provide some additional clarity on the procedures.
Table 1 - School Grade, can you help to clarify this? They are not in order and range from 3-22.
One flaw or concern is that most of the students in the study were from a rural area which may have changed the outcomes of the study. This should be highlighted in the discussion. You spend some time on this in lines 310-313 but I feel it is needs more connection to the literature.
Line 260: Add comma after "behavior"
Line 333: Add comma after "relatedness"
Line 347: I would provide some explanation or suggestions for the data regarding the overweight children. If I am interpreting correctly, this is only 7 participants and we need to be mindful of how this can be interpreted by the reader that if you have a normal weight child, it doesn't matter - which is not true from your data.
Line 344: add comma after "self-esteem"
Author Response
Reviewer 2
First, we would like to thank the reviewer for the careful examination of our manuscript. We sincerely appreciate the very precise comments. We have revised the manuscript accordingly.
Line 26: Add the year with the date
Amended as requested.
Line 43-44 (and throughout the rest of the paper): Why do you provide citations with names/years here but also in the requested format as brackets? I would suggest removing all named citations like this throughout the paper.
Amended as requested.
Line 79: Grammar issue with (during to after SaH). Please revise.
Amended as requested.
Line 123: Remove the semicolon and replace with 'or'. Semicolon would not be used in this instance
Amended as requested.
Line 142: The CDC title should be capitalized
Amended as requested.
Line 153-157: Not sure you need to list out the exact questions like this. I would suggest removing and provide the citation for the instrument.
Amended as requested.
Line 161: Can you provide more details regarding the 10 day data collection period? Was this consecutive days or only days when they had school? It also states "during school" - does this include online education if the schools were shut down? Please provide some additional clarity on the procedures.
Thank you for this comment. The devices were worn for 10 consecutive days during school hours and leisure time. We clarified this in the amended text.
Table 1 - School Grade, can you help to clarify this? They are not in order and range from 3-22.
Thanks for the attentive reading. The table has been corrected (see table 1).
One flaw or concern is that most of the students in the study were from a rural area which may have changed the outcomes of the study. This should be highlighted in the discussion. You spend some time on this in lines 310-313 but I feel it is needs more connection to the literature.
Thank you for this comment. We amended the limitations section to include this issue in further detail (see lines 369-378).
“Third, the sociodemographic background of the participants was not representative for Switzerland. Notably, PA data after SaH were found to be comparable to the SOPHYA study [62]. For example, similar values for time spent in MVPA were found in the SOPHYA (age groups 8-9 = 100.7 minutes; 10-11 = 72.5 minutes; 12-13 = 58.6 minutes) and the current study (age groups: 8-9 = 93.87 minutes; 10-11 = 69.61 minutes; 12-13 = 66.80 minutes). It nevertheless is possible that the effect of SaH on PA is underestimated as a previous study found largest reductions in urban areas and the sample of the current study consisted mainly of children from rural areas [30]. Given this selection bias, the generalizability of results to Swiss children in general is not possible.”
Line 260: Add comma after "behavior"
Amended as requested.
Line 333: Add comma after "relatedness"
Amended as requested.
Line 347: I would provide some explanation or suggestions for the data regarding the overweight children. If I am interpreting correctly, this is only 7 participants and we need to be mindful of how this can be interpreted by the reader that if you have a normal weight child, it doesn't matter - which is not true from your data.
Thank you for this comment. We rephrased this section and analyzed the correlations without the overweight children. When excluding overweight children from the correlational analyses, a significant relationship between BMI and change in HRQoL remains (e.g., for the scale of self-esteem). It therefore seems that relationships are also relevant for normal weight children. Because of the small sample size, we refrain from adding these additional analysis to the manuscript. We hope for your understanding.
“When having a closer look at the correlation (see figure S1 for scatter plot), it becomes apparent that overweight children have a particular influence on the correlation. We therefore speculate, that children with a larger BMI may have been particularly affected by SaH in terms of their psychosocial and physical HRQoL. However, this has to be interpreted cautiously because only seven children were overweight in the current sample.”
Line 344: add comma after "self-esteem"
Amended as requested.
Reviewer 3 Report
There are now many empirical studies of the impact of COVID19 restrictions on children's health and wellbeing. The present study provides an original contribution, particularly with the inclusion of a Quality of Life measure. The present study is interesting as a stand-alone piece of work, but I believe it will be more powerful when picked up for reviews of COVID restrictions that will no doubt occur. The study is appropriately designed and analysed. The researchers are very clear about the limitations of this study and avoid over-interpretation of the data.
This is a repeated measures design, not a longitudinal design. Studies with data collected at two time points are sometimes referred to as longitudinal, but this is incorrect. Longitudinal data are collected to examine changes over time and therefore need at least 3 data collection points to fit a line. The current study examines changes due to an 'intervention' and only has 2 data collection points, so it is a repeated measures design.
Were there any adjustments made for differences in accelerometer wear time for Time 1 and Time 2? Apologies if I missed this, but it seems as though the minutes in S,L,M,V were recorded without adjustment. If wear time is different for Times 1 & 2 this will have an impact. From Table 2 there were 991.56 minutes in Time 1 and 1004.61 in Time 2. Although a small difference, it could be important. I also noticed that the number of minutes reported for MVPA was slightly different to the number of minutes for Moderate + Vigorous. This was the case for Times 1 and 2. There may be an error that needs to be corrected.
Line 46: change 'to' to 'with'
Line 57: change findings to past tense: ...were worried, had higher levels of stress....
Author Response
First, we would like to thank the reviewer for the careful examination of our manuscript. We sincerely appreciate the positive feedback and the constructive comments. We have revised the manuscript accordingly and we are sure that it has improved by these changes.
There are now many empirical studies of the impact of COVID19 restrictions on children's health and wellbeing. The present study provides an original contribution, particularly with the inclusion of a Quality of Life measure. The present study is interesting as a stand-alone piece of work, but I believe it will be more powerful when picked up for reviews of COVID restrictions that will no doubt occur. The study is appropriately designed and analysed. The researchers are very clear about the limitations of this study and avoid over-interpretation of the data.
Thank you for your positive feedback.
This is a repeated measures design, not a longitudinal design. Studies with data collected at two time points are sometimes referred to as longitudinal, but this is incorrect. Longitudinal data are collected to examine changes over time and therefore need at least 3 data collection points to fit a line. The current study examines changes due to an 'intervention' and only has 2 data collection points, so it is a repeated measures design.
We really appreciate you taking the time to explain this to us! We are now using the term repeated measures design instead of longitudinal.
Were there any adjustments made for differences in accelerometer wear time for Time 1 and Time 2? Apologies if I missed this, but it seems as though the minutes in S,L,M,V were recorded without adjustment. If wear time is different for Times 1 & 2 this will have an impact. From Table 2 there were 991.56 minutes in Time 1 and 1004.61 in Time 2. Although a small difference, it could be important. I also noticed that the number of minutes reported for MVPA was slightly different to the number of minutes for Moderate + Vigorous. This was the case for Times 1 and 2. There may be an error that needs to be corrected.
Thank you for this thoughtful comment. We compared the difference in total wear time from Time 1 vs. Time 2 using paired sample t-tests. Because we did not find statistically significant differences (p > .05), the subsequent calculations were made without adjustment. This information was added to the statistical analyses section of the manuscript (see lines 190-194).
“Statistical tests were performed using SPSS 27.0 (SPSS Inc., Chicago, IL, USA). First, the total accelerometry wear time was compared between the first and second measurement point using paired sample t-tests. Because we did not find statistically significant differences (p > .05), unadjusted paired sample t-tests were used for the comparisons of PA and HRQoL during SaH with after SaH”.
The slight differences you recognized are due to the circumstance that the descriptive statistics include the averages from the multiply imputed datasets. To inform the attentive reader, we added this information to the manuscript:
“Note that descriptive statistics (e.g., in table 1) include the averages from the multiply imputed datasets.”
Line 46: change 'to' to 'with'
Amended as requested.
Line 57: change findings to past tense: ...were worried, had higher levels of stress....
Amended as requested.
Round 2
Reviewer 2 Report
None. Well done on the revisions.